Severe airport sanitarian control could slow down the spreading of COVID-19 pandemics in Brazil

Ribeiro Sérvio Pontes 1 2 3 spribeiro@ufop.edu.br
http://orcid.org/0000-0003-2863-8076 Castro e Silva Alcides 4
http://orcid.org/0000-0002-4758-4379 Dáttilo Wesley 5 wesley.dattilo@inecol.mx
Reis Alexandre Barbosa 1 6
http://orcid.org/0000-0002-7692-6243 Góes-Neto Aristóteles 7
http://orcid.org/0000-0002-6769-9931 Alcantara Luiz Carlos Junior 8 9
Giovanetti Marta 8
Coura-Vital Wendel 1 10
Fernandes Geraldo Wilson 11
http://orcid.org/0000-0002-4775-2280 Azevedo Vasco Ariston C. 9
1 Núcleo de Pesquisas em Ciências Biológicas, Universidade Federal de Ouro Preto , Ouro Preto, Minas Gerais , Brazil
2 Laboratório de Ecohealth, Ecologia de Insetos de Dossel e Sucessão Natural-ICEB, Universidade Federal de Ouro Preto , Ouro Preto, MG , Brazil
3 Laboratório de Fisiologia de Insetos Hematófagos-DEPAR, Universidade Federal de Minas Gerais , Belo Horizonte, MG , Brazil
4 Laboratório da Ciência da Complexidade-Departamento de Física, Universidade Federal de Ouro Preto , Ouro Preto, Minas Gerais , Brazil
5 Red de Ecoetología, Instituto de Ecología AC , Xalapa, Veracruz , Mexico
6 Laboratório de Imunopatologia-Departamento de Análises Clínicas-Escola de Farmácia, Universidade Federal de Ouro Preto , Ouro Preto, Minas Gerais , Brazil
7 Laboratório de Biologia Molecular e Computacional de Fungos-Departamento de Microbiologia-ICB, Universidade Federal de Minas Gerais , Belo Horizonte, Minas Gerais , Brazil
8 Laboratório de Flavivírus, Instituto Oswaldo Cruz, Fundação Oswaldo Cruz , Rio de Janeiro, Rio de Janeiro , Brazil
9 Laboratório de Genética Celular e Molecular-Departamento de Genética, Ecologia & Evolução-ICB, Universidade Federal de Minas Gerais , Belo Horizonte, MG , Brazil
10 Laboratório de Epidemiologia e Citologia-Departamento de Análises Clínicas-Escola de Farmácia, Universidade Federal de Ouro Preto , Ouro Preto, MG , Brazil
11 Departamento de Genética, Ecologia & Evolução/ICB, Universidade Federal de Minas Gerais , Belo Horizonte, Minas Gerais , Brazil
Folayan Morenike
Electronic publication date: 2020 Jun 25
Publication date: 2020
Volume: 8
Electronic Location ID: e9446
Received 2020 Mar 23; Accepted 2020 Jun 8
Copyright: © 2020 Ribeiro et al.
Copyright year: 2020
Copyright holder: Ribeiro et al.
License: This is an open access article distributed under the terms of the Creative Commons Attribution License, which permits unrestricted use, distribution, reproduction and adaptation in any medium and for any purpose provided that it is properly attributed. For attribution, the original author(s), title, publication source (PeerJ) and either DOI or URL of the article must be cited.
License URL: https://creativecommons.org/licenses/by/4.0/

Keywords: SIR model, Metapopulation dynamics, Amazonia, Indigenous people, One-Ecohealth, SARS-CoV-2 pandemic

Funding: CNPq Agency The CNPq agency guaranteed research grant scholarships for Sérvio Pontes Ribeiro, Alexandre Barbosa Reis, Aristóteles Góes-Neto, Luiz Carlos Junior Alcantara, Geraldo Wilson Fernandes and Vasco Ariston C Azevedo. The funders had no role in study design, data collection and analysis, decision to publish, or preparation of the manuscript.

==============================
Background

We investigated a likely scenario of COVID-19 spreading in Brazil through the complex airport network of the country, for the 90 days after the first national occurrence of the disease. After the confirmation of the first imported cases, the lack of a proper airport entrance control resulted in the infection spreading in a manner directly proportional to the amount of flights reaching each city, following the first occurrence of the virus coming from abroad.

Methodology

We developed a Susceptible-Infected-Recovered model divided in a metapopulation structure, where cities with airports were demes connected by the number of flights. Subsequently, we further explored the role of the Manaus airport for a rapid entrance of the pandemic into indigenous territories situated in remote places of the Amazon region.

Results

The expansion of the SARS-CoV-2 virus between cities was fast, directly proportional to the city closeness centrality within the Brazilian air transportation network. There was a clear pattern in the expansion of the pandemic, with a stiff exponential expansion of cases for all the cities. The more a city showed closeness centrality, the greater was its vulnerability to SARS-CoV-2.

Conclusions

We discussed the weak pandemic control performance of Brazil in comparison with other tropical, developing countries, namely India and Nigeria. Finally, we proposed measures for containing virus spreading taking into consideration the scenario of high poverty.

Introduction

The new disease COVID-19 has been spreading rapidly around the world since early January. It started in China at the end of 2019, been declared a “Public Health Emergency of International Concern” in 30 January and a pandemic in 11 March (World Health Organization (WHO), 2020a; Zhu et al., 2020). Its form of transmission is mainly through respiratory droplets of infected patients and contact with surfaces infected by aerosols (Fathlzadeh et al., 2020). However, the transmission dynamics has changed quickly in few months, with R0 varying from 0.3 to 2.0 in some countries and close to 3.0 in others (Lai et al., 2020a; World Health Organization (WHO), 2020a). Large continental countries are likely to be very vulnerable to the occurrence of pandemics (Morse et al., 2012; Dáttilo et al., 2020). While the dissemination dynamics have varied between regions, country sanitary policies play a key role. For instance, two very large developing countries, India and Brazil, have a very different epidemical pattern. On March 18th, India had 137 cases and Brazil 621, as recorded in the Brazilian Ministry of Health (https://covid.saude.gov.br/, 2020) and John Hopkins (https://gisanddata.maps.arcgis.com/, 2020) monitoring sites dedicated to SARS-CoV-2 and COVID-19. From 17th to 18th March, Brazil had an increase of 31% in 1 day, with only four capitals exhibiting community transmission, which was the same in India. Nonetheless, a very distinct pattern in the ascending starting point for the reported disease exponential curve was observed in each country. By enlarging the comparison to another developing tropical country in the Southern Hemisphere (thus, in the same season), we selected Nigeria since it was the first country to detect a COVID-19 case in sub-Saharan Africa (World Health Organization (WHO), 2020d). Nigeria reported eight confirmed cases during the same period of time (Nigeria Centre for Disease Control, 2020). Furthermore, Nigeria has a population similar to that of Brazil (201 million and 211.1 million, respectively, according to United Nations, 2019).

Both India (Airports Authority of India, 2020) and Nigeria (Federal Airports Authority of Nigeria, 2020) ensured severe entrance control, and close follow up of each confirmed case, as well as their living and working area, and people in contact with them, differently from Brazil. In Brazil, the Ministry of Health has developed a good monitoring network and a comprehensive preparation of the health system for the worst-case scenario. Nonetheless, apparently, the decisions from the Ministry of Health did not cover airport control, and only on March 19th, eventually too late, the government decided to limit the number of flights coming from Europe or Asia. Hence, the entrance of potentially exposed passengers to SARS-CoV-2 in Brazil has been occurring with no control, at least until the aforementioned date. Moreover, after confirming that a person is infected with SARS-CoV-2, his/her monitoring is initiated but there is no monitoring of potential contacts.

For pandemic situations, the classical algebraic ecological models of species population growth from Verhulst, and species interaction models from Lotka-Voterra, are theoretical frameworks capable to describe the phenomenon and to propose actions to stop it (Pianka, 2000). In many aspects, social distancing is a way to severely reduce carrying capacity, that is, the resources available for the virus dissemination. This is the best action for within-city pandemic spreading of this new coronavirus (Hellewell et al., 2020), since the main form of transmission is direct contact between people or by contact with fomite, mainly in closed environments, such as classrooms, offices, etc. (Rothe et al., 2020; Bedford et al., 2020). Regardless of virulence, for a highly contagious virus such as SARS-CoV-2, the occurrence of the first case in a nation will result in a strongly and nearly uncontrollable exponential growth curve, depending only on the number of encounters between infected and susceptible people, and fueled by a high R0, which is ranging from 0.8 to 3.6 for COVID-19, depending on region and period analyzed (Lai et al., 2020a, 2020b).

On the other hand, the dynamics of disease spreading among cities are entirely distinct. This is a growing scientific area, an obliged interdisciplinary field, where neural network driven epidemiological phenomena is central (Brockmann & Helbing, 2013); however, it must be defined by metapopulation and closeness, as well as by an individual gene flow driven phenomenon (Colizza & Vespignani, 2007). In order to understand subpopulation flow between demes—which are cities—connected by roads or air flight networks, several complex details may be raised in the process of modelling such as reality (Balcan et al., 2010). For instance, probabilistic motivated effective distance, which is the actual proximity caused by business, culture, and investments between two cities, is much more important than real geography, especially for airlines network (Brockmann & Helbing, 2013). Nevertheless, for such theoretical approach to infection disease dissemination, an ecological concept is key: metapopulation (Hanski, 1998). However, ecologists are those who are least devoted to explore human–pathogen interaction at a global or continental scale. The ecological approach may focus on the most biological aspect of these dynamics, which is simply virus dissemination inside an infected person. As typically designed in Ecology, the first attempt will be based on the most neutral model (sensu Hubbell, 2001), that is, the directions taken through the most frequent and intensively used links.

In this work, we present an epidemiological model describing the free entrance of people coming from two highly infected countries with close links to Brazil: Italy and Spain. We show how SARS-CoV-2 had spread into the Brazilian cities by the international airports, and then to other, less internationally connected cities, through the Brazilian airport network. For exploring the dynamics of a continent size, nationwide spreading of SARS-CoV-2, as it is the case of Brazil, we assumed cities connected by airports (simply cities hereafter) formed a metapopulation structure.

Each person in a city was taken as a component of a superorganism, that is, an interdependent entity where living individuals are not biologically independent between them in various subtle ways. By doing so, we dealt with cities as the sampling units, not the people, and, therefore, our model is slightly different from classic bolsonic models (Colizza & Vespignani, 2007; Nicolaides et al., 2012). Flights coming from foreign countries with COVID-19 (namely Spain and Italy for this article) represent the probability of an external introduction of infection in each city. Additionally, we also further explored the vulnerability of the Amazon region, especially of those remote towns where indigenous and traditional communities predominate.

Materials and Methods

In order to describe the pattern of air transportation and its role in the spreading of the disease, we built a Susceptible-Infected-Recovered (SIR) model (Hethcote, 1989; Anderson, 1991) split amongst the cities that are interconnected by flights. In this model, the population size inside each city is irrelevant. Moreover, the number of flights is highly correlated with city population (R2 = 0.76; p < 0.0001) and, thus, a good proxy of city size. Similarly, to our purpose the time when the collective infection stage was reached inside each city was irrelevant. Thus, we assumed that the city was fully infected and became infectious to the whole system, and, therefore, became a source and not a sink of infection events, after a certain amount of arriving contaminated people sums up. Hence, the SIR model started having cities with only susceptible events and change of these to the infected stage was counted as proportion of the population. Infected events only appeared by migration, that is, travelers only from Italy and Spain, for sake of simplicity and proximity to the early facts.

After the first occurrence having been recorded in the country, infected people started to spread through the national airlines, and this spreading is proportional to the amount of infected people accumulating in a city (see the model explanation below). Because of that, the model causes a transient timing, when the amount of infected people arriving is greater than the community transmission. Afterwards, the city started to disseminate to other places by probability rules. As the transmission is quite likely to happen by the simple proximity person-to-person, due to the resistance of the virus in air droplets and surfaces, as hands or metals, we assumed that a simple encounter will cause infection. However, this became an inner trait of each city that after having local transmission, starts to transmit to other cities as explained above. In addition, the model assumes that there is no public control in people circulation after arriving of an infected person. Ethical approval was not necessary as human participants were not involved in the study.

We used a modified version of the SIR model, which took into account the topology of how the cities-demes were linked by domestic flights. In the SIR original model, the infection of susceptible cities occurs by probability β of a healthy being (S) encounters an infected one (I). Conversely, the model has a probability of an infected one get recovered (R) given by a parameter γ. Analytically: St+1=St−βNStIt

It+1=It+βNStIt−γIt

Rt+1=Rt+γIt

where the indexes t and t + 1 represent the present time and the next time, respectively, and N = S + I + R is the total constant population. In this work, we proposed two modifications of the SIR model. The first one is related to the fact that we considered all the Brazilian cities that have an airport as subpopulations, meaning that each city has its own SIR variables set. Thus, we had Si, Ii, and Ri where i was a given city. In our case study, 1 ≤ i ≤ 154. The second important modification was related to the connections among the cities, which we used as a network of disease dissemination. This city network is given by the domestic flights among all the airports in Brazil, taking into account the number and the direction of flights. Therefore, we have a weighted network, where the weighted closeness centrality of each city was measured. Thus, the flights provided a dependent mechanism by which the between-cities connections caused a coupling of SIR equations based on the network flight structure. This coupling is provided by the last equation and is mediated by the new introduced parameter alpha. Using Agência Nacional de Aviação Civil (ANAC) (2020) data, it was possible to track all the domestic flights in Brazil (Fig. 1).

Figure 1 Brazilian flight network, taken from ANAC database.

The modified version of SIR model is then described as follows: St+1i=Sti−βNSti(Iti+Iti¯)

It+1i=Iti+βNSti(Iti+Iti¯)−γ(Iti+Iti¯)

Rt+1i=Rti+γ(Iti+Iti¯)

where the upper index i indicates the city, and t the time. The term Iti¯ represents the infection added to the ith city due to traveling diseases, and it is calculated as follow: Iti¯=α∑j=0154⁡kj,iIj

where kj,i is the number of flights departing at city j and arriving at city i, and α is a newly introduced parameter, which represents the fraction of traveling infected population. For the time, considering the geographic of Brazil and the time-lag for the dissemination towards the most remote cities, we estimated 90 days of disease expansion and assumed γ as 0, in other words, no recovery. Despite the artificiality of this assumption, we considered that the amount of people still to be infected is larger than those recovered and, thus, becoming resistant. For instance, by 19th May the recovery rates in Brazil reached 85%, but this corresponded to only 100.5 thousand people around the country (www.worldometers.info/coronavirus/country/brazil/, 2020). Divided between each infected city, this makes mathematically small numbers, and thus resistance becomes demographically irrelevant to our output of early disease dissemination. Furthermore, by dealing with only the infected portion of the population (proportion of infected), we avoided the uncertainties related to the little known COVID-19 resistance development (Li et al., 2020).

The model was developed in C and is available as Supplemental Material 1 (and the database as Supplemental Material 2). In addition, we also used a linear model to test whether those cities with higher city closeness centrality (i.e., important cities for connecting different cities within the Brazilian air transportation network) were more vulnerable to SARS-CoV-2 dissemination. The Beta parameter was defined by calibrating the model with real time series from Johns Hopkins Coronavirus Research Center (https://coronavirus.jhu.edu/map.html), which leads to β = 0.3035 (Supplemental Material 1). We used the value α = 0.0001 based on the amount of flights needed from the peak of the disease in Italy until an infected person was recorded from that country in Brazil.

Results

The expansion of the SARS-CoV-2 virus between cities was fast, directly proportional to the city closeness centrality within the Brazilian air transportation network. The disease spread from São Paulo and Rio de Janeiro to the next node-city by the flight network, and, in 90 days, virtually all the cities with airport(s) were reached; however, it occurred with a distinct intensity (Fig. 2; Supplemental Material 3). There was a clear pattern in the expansion of the pandemic, with a stiff exponential expansion of cases (measured as the cumulative percentage of infected people per city) for all the cities. On average, the model showed an ascendant curve starting at day 50 (around 15 April), with the most connected cities starting their ascendant curve just after 25 days, and the most isolated ones from day 75 (10th May; Fig. 3A). Looking at the daily increment rates, it is clear a fast and high peak of infections in the hub cities, happening around 50 days and, starting from 75 days, a new peripheric peak (Fig. 3B).

Figure 2 Proportion of infected population of each Brazilian city in 40 (A), 50 (B), 70 (C), and 90 (D) days.

Circle colour temperature represents a gradient in percentage of the infected population. Circle size also reflects the size of the pandemics locally in the logarithm scale.

Figure 3 Proportion of infected people per cities until 90 days.

(A) Cumulative increment rate. The blue line is the national average, and the shadow area is the summing up of minimum and maximum values of all the cities per time interval; (B) daily increment rate. The blue line is the average, showing the overall high rate of infection occurring from 50 to 80 days. Shadow shows the first and the highest peak in the hub cities, around 50 days, and, subsequently, a peripheric peak after 75 days.

The first ten cities to ascend infection rates (São Paulo, Rio de Janeiro, Salvador, Recife, Brasília, Fortaleza, Belo Horizonte, Porto Alegre, Curitiba, and Florianópolis) will actually reach this point about the same time, which is a concerning pattern for the saturation of the public health services. Moreover, this peak in those cities will saturate all the best hospitals in the country simultaneously.

Therefore, we defined the average proportion of infected people for the 90 days as a measure of the vulnerability to COVID-19 dissemination. Henceforth, we found the more a city shows closeness centrality within the air transportation network, the greater was its vulnerability to disease transmission (Fig. 4). This scenario confirmed the importance of a city connecting different cities within the Brazilian air transportation network and, thus, acting as the main driver for the pandemic spreading across the country.

Figure 4 Airport closeness centrality within the Brazilian air transportation network, and its effect on the vulnerability of each city.

Correlation between airport closeness centrality within the Brazilian air transportation network, and its effect on the vulnerability of each city (represented by the average of the percentage of cases per city for the whole 90 days running: r2 = 0.71 p < 0.00001).

Consequences for the Amazonian cities and indigenous people

Herein, we showed that an uncontrolled complex airport system made a whole country vulnerable in few weeks, allowing the virus to reach the most distant and remote places, in the most pessimistic scenario. According to our model, any connected city will be infected after 3 months. As the number of flights arriving in a city is the driver for the proportion of infected people, Manaus, which is a relevant regional clustering, was infected sooner. Indeed, on the 17th of March, Manaus was the first Amazonian city with confirmed cases (without community transmission yet, according to the Ministry of Health: https://covid.saude.gov.br/, 2020), and it is a node that is one or two steps to all the Amazonian cities. Thus, according to our model, Manaus may reach 1% of the infected population by the 44th day, while, for instance, the far west Amazonian Tabatinga will take 61 days to reach the same 1% of the population infected. By day 60, Manaus may have an average of 50% of its population infected if nothing is done to prevent it. Tabatinga may also reach the aforementioned value by day 78, if nothing is done to avoid it. To sum up, within 46 days, all the Amazonian cities will have 1% of their population infected and a mean of 50% by day 70.

Discussion

Our model suggests that Brazil must be prepared for an exponential rise in COVID-19 cases within 3 months from end of February, starting synchronously by the wealthiest cities. Such increase is expected based only on the dissemination among cities by the commercial airports and may get worse without the measures of social distancing proposed by the WHO (Coelho et al., 2020). The Country has failed to contain COVID-19 in airports and to closely monitor those infected people coming from abroad, as well as their living network. According to the Brazilian Airport Authority (Agência Nacional de Aviação Civil (ANAC), 2020), Brazil has the second-largest flight network in the world (just after the USA), with a total of 154 airports registered to commercial flights of which 31 are considered international. In comparison, airport control may be much easier to set up in Nigeria (31 airports of which only five are international: Federal Airport Authority of Nigeria, 2020). Nonetheless, with a population 6.4 times higher than Brazil (United Nations 1919), India, in turn, has a similar sized airport network to Brazil, harboring a total of 123 airports of which 34 are considered international (Airports Authority of India, 2020).

Nevertheless, the situation of COVID-19 in India is currently much milder than in Brazil, and it is hard to blame the complexity of the airport networks for the contrasting exponential curve of these two countries. In 20 days, from the first infection in Brazil (February 26th) against 47 days after the first Indian case (January 30th), Brazil has already had 5.4 more confirmed cases than India (https://covid.saude.gov.br/, 2020; https://gisanddata.maps.arcgis.com/, 2020). Clearly, one country is doing much better in preventing the entrance of cases and the spreading of the disease by controlling infected citizens. Indeed, according to the WHO, the Ministry of Health and Family Welfare (MoHFW) of India has taken an early action, and “aggressively stepped up the response measures—find, isolate, test, treat and trace” (World Health Organization (WHO), 2020b).

Nigeria, a country poorer (27th world GDP position with $446,543 billions nominal) than Brazil (9th world GDP position with $1.85 trillion nominal) but with a similar population (United Nations, 2019), had imposed a very successful control so far, with guidelines and laws for within all the cities embracing social distancing and obliging wearing of face masks, starting from 31st March, 2 weeks before than Brazil (Nigeria Centre for Disease Control, 2020). As in most of the African countries, screening at the points of entries have been conducted in Nigeria’s airports (World Health Organization (WHO), 2020c). The Federal Airports Authority of Nigeria (2020) had established precautionary measures to be observed by inbound passengers, the opposite that Brazilian authorities did (Agência Nacional de Aviação Civil (ANAC), 2020).

In order to find and isolate is, from an ecological perspective, the most efficient way to reduce the carrying capacity of a new disease, and thus, restrict its wide spreading, and this must start at the airports. As Brazil is just struggling to impose social distancing, a State-to-State decision with little support from the Federal Government, the scenario is evolving more severely. Regardless of flaws in the comparisons of confirmed cases between the countries, by 16th April, India still had 12,380 cases and 414 deaths while Brazil had 25,262 cases and 1,532 deaths (World Health Organization (WHO), 2020c). Nigeria has 373 cases and 11 deaths (World Health Organization (WHO), 2020c). Despite the clearly more severe airport sanitarian control in Nigeria than in Brazil, one needs to be aware of the unexpectedly low number of cases in the continent, which may be related to a very young population or to other situations not well understood yet (Vaughan, 2020). For Brazil, on the contrary, enhanced case detection would make the scenario even worse. Considering the high probability of a synchronizing SARS-CoV-2 spreading in various capitals, the country may face a quick health service collapse.

Besides the within-city pattern of virus spreading, one must take into account the pattern of dispersion between cities after the virus has invaded. Additionally, for the Brazilian case, one cannot ignore that, eventually, the occurrence of the first case may have occurred nearly 1 month before official records, during the carnival period. This is the largest popular street party on the planet, with 6.4 million people in Rio de Janeiro, and 16.3 million in Salvador, and the Brazilian Ministry of Tourism (2020) revealed that 86,000 foreigners from France, Germany, Spain, Italy, UK, and the US had visited Brazil in this period. As airport control might have been even more lax in small airports, it might unavoidably result in strengthening of the capability of an infected city to infect the next new one, if no public policy is adopted.

Without a social distancing policy, virus propagation may result in chaotic dynamics, sensu May (1976). The lack of control for these situations may result in a dramatic rate of host infection, and an eventual collapse of the host-parasite interaction in a given population, depending on the amount of susceptible, infected, and recovered events. Nonetheless, if the population is split into deme-cities, in a metapopulation structure, the collapse takes longer, and a much greater amount of people in different locations may eventually be infected, as found in our model. It is worthwhile to mention that this model, already pessimistic, did not consider the Brazilian road network, one of the largest on the planet. Most importantly, the best road-connected cities are exactly those mostly connected by airport, and that will be vulnerable earlier, and thus, probably spreading the disease faster than our model can predict, unless roads are soon blocked for people. Another weakness of the model is that it cannot account for a great number of small airports not registered for commercial flights, very common in the Amazonian and Western regions. Taking this into a global scale, for a highly interconnected human population, the consequences may be catastrophic, as it was for the influenza pandemic (Spanish flu) in 1918 (Fergunson, Alison & Bush, 2003). Furthermore, one aspect that must not be neglected is the way an increasing number of infected people in a city drives the pandemic towards the next city or country. In this context, the complex and large flight network of Brazil, which is also key for the whole Latin America, if not properly monitored and controlled, may cause a window of opportunity for the virus to spread over the entire continent.

The consequences of this uncontrolled SARS-CoV-2 spreading is particularly serious if one takes into consideration the chances of a mutant virulent strain appearing and spreading into poorer and little monitored places of the world. Specifically, for the Amazon region, the lack of any control will make the city of Manaus a very sensitive cluster for public health, due to predominantly poor and indigenous-dominated cities in the region, which are connected to Manaus and will be rapidly infected. Reaching isolated regions means reaching indigenous or traditional communities, whose individuals are classically more susceptible to new pathogens than western-influenced or mixed urban populations. Therefore, a way to prevent such spreading, if still there is time, would be to deal with airports as entrances that need severe infection barriers.

Conclusions

The Brazilian media and the Ministry of Health have announced that 50 days from the introduction of SARS-CoV-2 in the country, it has spread across the regions and cities quite like we had predicted, even slightly faster. For instance, the time taken for Manaus to be compromised by the disease was as short as we predicted. A combination of being an important regional clustering in the airport network, and relatively limited hospital capability, resulted in a fast saturation of intensive care units, in cities as Manaus, Recife, and São Luis. As Manaus will disseminate the disease across the region, quickly reaching far remote indigenous communities, it should be on lockdown right now, to cause an invasion threshold, sensu Colizza & Vespignani (2007). The first indigenous person, a teenager girl, has been already killed by COVID-19.

An eventual lesson to take for the whole country is that inflexible, severe, and easy to repeat protocols must be applied to all the cities with airports. Likewise, the follow-up monitoring of suspicious individuals and their living network should be reinforced as a national strategy to prevent a large territory to be taken over by a pandemic in a short period of time. In other words, internationally accepted procedures must be taken and even be reviewed to adjust to complex national flight networks of any country. Such procedures must be considered as a priority for national remote airports too, in order to keep poorer and worse equipped cities away from a rapid spread of a pandemic disease.

It is clear, at this point, that a fast spread of the SARS-CoV-2 is a reality in Brazil, and across most of the country. We proposed this model in order to emphasize the fragility of Brazilian surveillance in the airport network, in an attempt to cause some policy change in time to preserve at least the most remote regions, which are also the most vulnerable, with a weaker health service. Moreover, most of the Eastern part of the country must stay in social distancing in order to prevent a health public collapse by mid-May, as the Brazilian Ministry of Health predicted. So far, this has been a State-to-State decision, similarly to the United States, and those States that have been stronger in isolation measures, are indeed delaying the peak. Moreover, even if it is too late for the first wave of infection in many cities, one must be prepared for a second likely wave, mainly considering the lack of a central government policy for social distancing.

In addition, we also could consider the generalized poverty of Brazil as a further problem that our model did not deal with. The chances to produce home-to-home isolation, even legally imposed, is impossible for these poor communities. Nonetheless, considering the few main entrances of most of the Brazilian shanty towns and communities, a similar to airport entrance severe control must be considered to protect a larger but closely connected set of people, eventually following the protocols used for the control of Ebola during the last epidemic in Africa (Lau et al., 2017).

Supplemental Information

Supplemental Information 1 Code description - SIR model under network topology.

The code was developed in C, and it works as a modification of SIR model running along with the topology of the domestic flight network. After initiating all variables to an initial condition, that is, S (health), I (infected) and R (recovered) of each city, the code starts loading the network and calculates the total number of flights among all the cities. This information is used to feed the classical SIR model introducing in the variable I, the information regarding infected travelers and non-travelers, and the model calculates the next S, I and R of all the cities. This calculation is done in a loop time representing days, the time step that the model was calibrated.

Click here for additional data file.

Supplemental Information 2 ANAC database of aerial transportation network.

The spreadsheet presents all the 120 cities with airport(s), their state, latitude and longitude, followed by the closeness centrality in the network. The columns t0 to t90 are the times from 0 to 90 days. Lines for the time columns are the percentage of infected people per city per time.

Click here for additional data file.

Supplemental Information 3 Movie of the spreading of SARS-CoV-2.

This file has a short movie describing the dynamics of SARS-Cov-2 dissemination across Brazil, in two versions.

Click here for additional data file.

We thank Christina Vinson and Thomas C.A. Williams for the English revision. We thank Muhammed Afolabi and two anonymous referees that contributed enormously to the improvement of this article.

Additional Information and Declarations

Competing Interests

Author Contributions

Data Availability

Aristóteles Góes-Neto and Vasco Ariston C Azevedo are Academic Editors for PeerJ.

Sérvio Pontes Ribeiro conceived and designed the experiments, performed the experiments, analyzed the data, prepared figures and/or tables, authored or reviewed drafts of the paper, and approved the final draft.

Alcides Castro e Silva conceived and designed the experiments, performed the experiments, analyzed the data, prepared figures and/or tables, authored or reviewed drafts of the paper, and approved the final draft.

Wesley Dáttilo conceived and designed the experiments, performed the experiments, analyzed the data, prepared figures and/or tables, authored or reviewed drafts of the paper, and approved the final draft.

Alexandre Barbosa Reis conceived and designed the experiments, authored or reviewed drafts of the paper, and approved the final draft.

Aristóteles Góes-Neto analyzed the data, authored or reviewed drafts of the paper, and approved the final draft.

Luiz Carlos Junior Alcantara analyzed the data, authored or reviewed drafts of the paper, and approved the final draft.

Marta Giovanetti analyzed the data, authored or reviewed drafts of the paper, and approved the final draft.

Wendel Coura-Vital conceived and designed the experiments, analyzed the data, authored or reviewed drafts of the paper, and approved the final draft.

Geraldo Wilson Fernandes conceived and designed the experiments, authored or reviewed drafts of the paper, and approved the final draft.

Vasco Ariston C Azevedo analyzed the data, authored or reviewed drafts of the paper, and approved the final draft.

The following information was supplied regarding data availability:

Raw data and code are available in the Supplemental Files.

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
