# Peer review of "Severe airport sanitarian control could slow down the spreading of COVID-19 pandemics in Brazil"

_PeerJ, doi:10.7717/peerj.9446_

## Round 0.1 · original submission · Major Revisions

Thanks for submitting your article to Peer Review

We look forward to your submitting the revised edition promptly.

·

Basic reporting

1. The manuscript was written in clear and unambiguous, professional English used throughout but authors should correct syntax errors in the following part of the manuscript:
a) Line 49: The use of 'diseased people' is generic in the context which authors used it. Authors should clarify if this description encompasses people with other transmissible diseases such as HIV, Hepatitis B virus etc, as they also come under the category of 'diseased people' as used by the authors.
b) Lines 162-163: '......where we found that....more an airport shows closeness centrality.....'

2. Although, authors provided sufficient introduction and background to demonstrate how the work fits into the broader field of knowledge. However, various statements in the introduction were not referenced. Authors should provide appropriate references to the following sections highlighted in the reviewed manuscript:

a) Line 41: Nigeria has a population of 206 million, similar to that of Brazil (209 million)

b) Lines 42-46: The use of 'claim' in the sentence sounds judgmental and may imply that the authors had pre-conceived bias as they did not believe the actions reported by airport authorities in India and Nigeria, but believed the Brazillian authorities. This becomes relevant because the findings of the study contradict the position/views of the authors. Authors should consider backing up this statement with appropriate references.

c) Lines 306-312: Authors should cite these URL with appropriate references properly, including the accessed dates

Experimental design

1. The article described original primary research which aligned with the scope of the journal. Authors should include in relevant section of the manuscript that 'ethical approval was not obtained or required because human participants were not involved in the study'.

2. Although the analysis reflected acceptable scientific rigour, the assumptions made in modifying the SIR model would benefit further explanation and justification:
a) Lines 94-95: This statement is not consistent with known aetio-pathogenesis of COVID-19 transmission. Usually, the index case or first occurrence is imported into the country via air travel by a person who visited an affected area. When the index case arrives their destination, transmission occurs by air droplets from the index case via coughing, sneezing, touching the infected surface areas, social interactions such hand-shaking etc. Authors should kindly check the following references and consider revising the assumption on which the modified SIR model was based: https://www.cdc.gov/coronavirus/2019-ncov/prepare/transmission.html;
https://www.eurosurveillance.org/content/10.2807/1560-7917.ES.2020.25.8.2000097

b) Lines 109-110: Authors described the first modification of the two modification of SIR model, but it is difficult to locate the description of the second modification in this section. Authors should clarify if the linear model described in the method section is the second modification of SIR model. If not, authors should describe the second modification clearly.

3. Line 39: This statement is debatable because COVID-19 was first reported in Egypt on 14 Feb 2020, ahead of Nigeria which was reported on 27 Feb 2020. Author should consider revising the sentence to read that Nigeria was the first country in sub-Saharan Africa to detect COVID-19.

4. Line 133: Assumptions in mathematical modelling usually mirror realistic estimates as much as possible. In real life, disease expansion for a highly contagious SARS-nCoV2 is much shorter than 90 days. For the conclusions of this study to benefit and shape public health policy, authors should consider re-visiting the estimated disease expansion or provide further justification to use 90 days in the context of a very rapidly expanding COVID-19.

5. Lines 135-136: This hypothesis has been recently disproved because about 14% of people who recovered from COVID-19 still carry the virus and can potentially infect other people. This does not support the theory of resistance being argued by the authors.
Authors should kindly refer to these reports and revise the hypothesis accordingly: https://jamanetwork.com/journals/jama/fullarticle/2762452
https://onlinelibrary.wiley.com/doi/full/10.1002/jmv.25685
https://www.forbes.com/sites/brucelee/2020/03/15/can-you-get-infected-by-coronavirus-twice-how-does-covid-19-immunity-work/#6b0f9e115c0f

Validity of the findings

The criteria for this section were satisfactorily met

Reviewer 2 ·

Basic reporting

Please see the review.pdf attached

Experimental design

Please see the review.pdf attached

Validity of the findings

Please see the review.pdf attached

Annotated reviews are not available for download in order to protect the identity of reviewers who chose to remain anonymous.

Reviewer 3 ·

Basic reporting

It would be great for the authors to highlight any existing measures already put in place at the airports and the gaps identified which then should form the basis of their model.

Line 26 and 27: there is need for a reference and suggestions below for the authors to consider; Novel Coronavirus (2019-nCoV) SITUATION REPORT - 1 21 JANUARY 2020
https://www.who.int/docs/default-source/coronaviruse/situation-reports/20200121-sitrep-1-2019-ncov.pdf?sfvrsn=20a99c10_4 and A Novel Coronavirus from Patients with Pneumonia in China, 2019 https://www.nejm.org/doi/pdf/10.1056/NEJMoa2001017?articleTools=true

Lines 30-33; There is need for a reference as well since the authors are highlight data that should be publicly available. By not citing, they make it difficult for the reader to try and look it up.

Line 39: This statement is inaccurate; “Nigeria, since it was the first country to detect a COVID-19 case in Africa” The following is what is correct and the authors should consider revising it and I have included references for the same. Nigeria was the 1st country in sub-Saharan Africa to report a COVID19 confirmed case but third in the entire continent after Egypt and Algeria. COVID-19, Situation 1 update for WHO African Region https://apps.who.int/iris/bitstream/handle/10665/331330/SITREP_COVID-19_WHOAFRO_20200304-eng.pdf and Coronavirus disease 2019 (COVID-19) Situation Report – 26 https://www.who.int/docs/default-source/coronaviruse/situation-reports/20200215-sitrep-26-covid-19.pdf?sfvrsn=a4cc6787_2

Line 40; For this statement, “Nigeria displayed less than 10 confirmed cases during the same period of time”, cconsider referencing this Nigeria Centres for Disease Control link for that information. https://covid19.ncdc.gov.ng/

Line 47-49 : “For this statement the government decided to control the airports, avoiding the entrance of people coming from Europe or Asia” By writing “avoiding” do you mean do you mean entry of passengers was stopped from these 2 continents? If so, please consider changing the word to ‘restrict’.

In line 48: What are these control measures? Were any existing before. May be good to give the reader an idea of what was done.

Line 49: Consider changing this statement “Hence, the entrance of diseased Brazil has been occurring” to “Hence, the entrance of potentially exposed passengers to SAR-COV-2 to Brazil has been occurring’. As written, it denotes occurrence of the disease withing the country, yet I believe the message being conveyed is the arrival of passengers who may be already exposed to COVID-19

Line 52: Consider changing “his/her living networks” to ‘potential contacts’ to underlie that monitoring needs to happen at the very of the case and contacts which is crucial in determining transmission dynamics and to put in place measures to disrupt transmission or contain the disease.

Line 53: Consider deleting “that with which were are living with”. It is confusing and without it the message remains unchanged and is clearer.

Line 79: Consider the word ‘introduction’ instead of the used one “invasion”

Experimental design

154: The authors meant from the statement ‘fast’ instead of what is written “first”. Kindly correct.

162” Add the word ‘the’ to make the statement complete.

175: There needs to be a reference of where you obtained the information about Manuas being the 1st Amazonian city with confirmed cases as of 17th March.

Validity of the findings

It would have been a good idea to summarise the model results in the first statement before stating the statement in 186. The message really is that there will be an exponential rise in cases by more than 50% within 3 months perpetuated by lack of intervention measures at the airports.

189-195: The authors need to add the reference. It makes it easy for a reader to look it up and these are data that has been generated by the relevant instructions thus should be available.

Line 199-203: It may be helpful to give the specifics of what India was doing different to Brazil for their case count to be low as described in the statement. What is written is general thus does not provide the reason why India was deemed to manage the situation better than Brazil.

Line 211-213: Is it really that majority were asymptomatic, or they have entered the country at the early stages of incubation period even before the latency period was over? Kindly look at these manuscripts below talking about CODID19 asymptomatic carriage. Asymptomatic carrier state, acute respiratory disease, and pneumonia due to severe acute respiratory syndrome coronavirus 2 (SARS-CoV-2): Facts and myths. https://www.ncbi.nlm.nih.gov/pubmed/32173241 and Presumed Asymptomatic Carrier Transmission of COVID-19 https://jamanetwork.com/journals/jama/fullarticle/2762028

Line 227: Consider using ‘account for’ instead of “quantify” so that the statement reads; “Another weakness of the model is that it cannot account for a great number of small airports not registered for commercial flights”.
Line 232: Delete “as”.

Figure 1. It may be helpful to show the main international airport as a reference point for the reader who is not conversant with the region. In line 146 reference is made to São Paulo and Rio de Janeiro airports as main hubs. This could be indicated in the figure.

Additional comments

Thank you for the opportunity to review the paper

This is a great paper trying to project the magnitude of COVID19 in country following introduction through air transport. It is a vital piece of analysis that can help policy makers and those in charge of the Response team to make timely and objective decisions. Thank you for putting it together. Find my comments below.

There is a lot of factual information cited but not referenced. Kindly ensure this is done. I have tried to include references in the specific comments for your consideration and feel free to use others sources that you are comfortable with.

Different fonts and size used in the narrative. i would imagine this would be handled during when setting the manuscript for publishing if approved.

The authors use Nigeria and India as comparison for the measures put in place but they do not mention specifically what was done in those countries and further in the discussion provide different scenarios of the outcomes if some of these measures are put in place.

The reference list in line 306-312 should be inline where cited. As current structured, it is a bit confusing and time consuming for one to figure out where they are used.

In the conclusion in the first paragraph, it may be good for the authors to provide specifics of the measures that need to be undertaken and how that relates to the model.

---

## Round 0.2 · Minor Revisions

Please address the remaining reviewer comments.

·

Basic reporting

Satisfactory

Experimental design

Satisfactory

Validity of the findings

Satisfactory

Additional comments

Authors have addressed concerns raised by the reviewer and made necessary corrections. Nevertheless, authors are encouraged to correct the following items and the Editorial team should kindly confirm these are done before publication of the manuscript:

Line 26: The opening statement (In the last few weeks) is currently not consistent with time evolution of COVID-19 pandemic. Authors should consider revising the phrase to reflect the COVID-19 timeline.

Line 27: 'Stealth transmission' has ceased to be the main route of COVID-19 transmission. Despite strict lockdown and restrictions placed on air travels in most parts of the world, COVID-19 transmission has not only remained unabated but is increasing at an exponential rate with Ro greater than 1 in several parts of the world. Authors should consider revising this statement to reflect the current transmission dynamics of SARS-CoV-2

Line 267: Author should consider supporting the sentence (Nigeria, a country poorer than Brazil) with a reference

Reviewer 2 ·

Basic reporting

In this paper, the authors investigate a scenario of COVID19 spreading in Brazil through the air-transportation network for 90-days after the first incidence. They use an SIR compartmental model where the country is divided into subpopulations served by airports. They find that closeness centrality of the different airports is what drives the epidemic spreading.

I find the authors edits after the reviews sufficient and therefore I believe that the study can be accepted for publication in PeerJ. I would suggest that the authors reconsider doing the following before the very final acceptance.

Major:

The authors believe that the size of the subpopulation does play a very minimal role in their model. However I believe it is misleading having in a model same population for Rio Airport and a small regional airport. If you see the equations you are evolving in time, the subpopulation size N appears in the denominator and therefore plays a role. If the authors consider running a robustness check with heterogeneous population size across Brasil, this will definitely strengthen the paper!

Minor:

There are some references appear in the text but not in the reference list. Please double check.

Experimental design

All my comments are above

Validity of the findings

All my comments are above

Reviewer 3 ·

Basic reporting

Line 28. Put a semi colon between the first and second references
Line 41. Consider replacing “displayed exactly” with “reported”.
Line 44 the United Nations 2019 reference is ambiguous. As currently stated, one will take time to find it. Kindly make it specific. Refer to the journal reference guidelines.
Line 46. ‘Ensured’ instead of ‘ensure’.
Line 51-52. For clarity you could consider changing the following statement “the government decided to impose 52 some limiting in the number of flights coming from Europe or Asia.” to “the government decided to limit the number of flights coming from Europe or Asia.
Line 60. Consider changing “social isolation” to “social distancing”
Line 63. Is the main form of transmission through direct contact or droplet infection? Of course, I am not negating the role of direct contact.
Line 69. Do you want to consider stating the current estimated R0 with a corresponding reference? Some papers here for an idea. Ranges from 1 to 3.5 depending on the literature one looks at and time period. Early phylogenetic estimate of the effective reproduction number of SARS-CoV-2. https://www.ncbi.nlm.nih.gov/pubmed/?term=Early+phylogenetic+estimate+of+the+effective+reproduction+number+of+SARS%E2%80%90CoV%E2%80%902
Severe acute respiratory syndrome coronavirus 2 (SARS-CoV-2) and coronavirus disease-2019 (COVID-19): The epidemic and the challenges. https://www.ncbi.nlm.nih.gov/pubmed/32081636
Line 90. Is it meant to be “showed” or “show” since this is still in the introduction section and is part of what your paper seeks to establish through the model?

Experimental design

A speculative question; Why `γ' presumed to be '0' given that recoveries are expected even from the already available of more than 60% recovering from SARS-COV-2 infection? Brazil, Out of 233 142 about more than 100, 000(42.9%) of the confirmed cases have recovered. https://www.worldometers.info/coronavirus/country/brazil/; https://www.who.int/docs/default-source/coronaviruse/situation-reports/20200518-covid-19-sitrep-119.pdf?sfvrsn=4bd9de25_4) Of course your next statement points to the artificiality of the statement

Validity of the findings

Line 189-195 in the first manuscript and now 241-247 in the revised one, the authors, in their rebuttal wrote they have included the reference. However, I could not see it. Could they confirm it is referenced and did not include it?
Line 235 change “raise” to “rise”.
Line 258. Than as currently described, “Nigeria, a country poorer than Brazil but with a similar population” This comes out as a bit subjective, would you consider using a measurable indicator? E.g GDP, infrastructure etc.?
Line 260. Add “wearing of” before “face masks”.
Line 263-265. The statement is unclear. I guess the message you are passing here is that the Nigerian authorities had in place precautionary measures to be adhered to by inbound passengers. If so, kindly rephrase the statement for clarity.
Line 269. Consider changing “drop down” to “reduce” and “refrains” to “restrict”.
Line 280. Consider changing “a more precise counting” to “enhanced case detection”. This is because the message you are passing here is increasing the ability to determine COVID19 cases.
Line 293. “Social isolation” or “Social distancing”. Given that “isolation” is largely used to for confirmed cases.
Line 327. Change “distributed” to “spread”.
Line 350. Comment similar to the one in Line 293.

Additional comments

Thank you Prof. Aristoteles and team for taking time to respond to the issue raised which only makes the manuscript better and enhances its veracity.
Find above few comments to address.
The conclusion seemed a bit long and cold be shortened.
The Lines references noted above are based on the version which i will attach as well with a few comments in it.
All the best.

Annotated reviews are not available for download in order to protect the identity of reviewers who chose to remain anonymous.

---

## Round 0.3 · Minor Revisions

There are only a few minor issues remaining, from Reviewer 3. I am looking forward to you re-submitting soon

·

Basic reporting

Satisfactory

Experimental design

Satisfactory

Validity of the findings

Satisfactory

Additional comments

I am pleased to confirm that authors have addressed the concerns I raised in my previous reviews.

Reviewer 3 ·

Basic reporting

Satisfactory

Experimental design

Satisfactory

Validity of the findings

Line 241-242: Delete ‘be’ “in this sentence “…….if nothing is be done to avoid it….” so that it reads ‘if nothing is done to avoid it’
Line 276-280: “The Federal Airports Authority of Nigeria (2020) have strict instructions for travelers in their page, from how to be conducted in front of the precautionary measures in the entrance to the hygienic behavior. The equivalent Brazilian agency just says “there is currently no restrictions for domestic or international flights” (ANAC 2020)” This was raised in the second review in which then it in line ‘Line 263-265” and the reviewer comment was: “The statement is unclear. I guess the message you are passing here is that the Nigerian authorities had in place precautionary measures to be adhered to by inbound passengers. If so, kindly rephrase the statement for clarity”. In their rebuttal, “Exactly, the sentence was re-written for sake of clarity” the authors stated they have addressed it but it still remains as in previous versions. For clarity it may be good to rephrase the sentence.

Line 287- 289. You may consider choosing the data from WHO or Nigeria CDC instead of having both. The variations in the data is because of when Nigeria CDC reports to WHO. In essence it is same data hence can opt for one of them whichever is latest by the of referencing.

Line 294: Change “SARS-Cov” to ‘SARS-C0V-2’


Lines 427 and 529; same comment as in line 294.
Line 346-347 Add “on” between ‘be’ and ‘lockdown’ “it should be lockdown right now”.

Additional comments

Dear Prof. Aristoteles and team, tyou have made great strides in addressing the comments raised by the reviewers. Minor comments from me.

Thank you

All the best

---

## Round 0.4 · Minor Revisions

Looking forward to the minor edits

Reviewer 3 ·

Basic reporting

Could the word ‘pandemics’ in the title and in line 28 be changed to ‘pandemic’?

Experimental design

Satisfactory

Validity of the findings

Line 292: ‘o’ is missing in ‘SARS-CV-2. It seems to have been deleted by mistake by the authors in the tracked changes.

Additional comments

Dear Prof. Aristoteles and team, you have made great strides in addressing the comments raised by the reviewers and the manuscript it is at stage to be considered for publication decision by the journal. Two errors noted above likely made during the editing process. It was a privilege to review your work.

One more general comment;

Line 378: Removes ‘s’ from “We thanks”

Thank you

All the best

---

## Round 0.5 · accepted · Accept

Thanks for making the needed changes